# Study on the Intercropping Mechanism and Seeding Improvement of the Cavity Planter with Vertical Insertion Using DEM-MBD Coupling Method

Linrong Shi [1], Wuyun Zhao [1,*], Chengting Hua [2], Gang Rao [1], Junhai Guo [1] and Zun Wang [1]

1 College of Mechanical and Electrical Engineering, Gansu Agricultural University, Lanzhou 730070, China
2 Hangzhou Nichuang Technology Co., Ltd., Hangzhou 310000, China
* Correspondence: zhaowy@gsau.edu.cn

**Abstract:** In the dry areas of Northwest China, cavity planters with vertical insertions are used for seeding on film. Due to the uncertain mechanism between cavity planters and maize seeds and soil, research on the cavity planter has been slow. Several theoretical and experimental methods have been developed to investigate the interaction between the cavity planter and maize seeds in soil. These methods enable exploration of the mechanism to reduce soil disturbance and improve seeding performance. However, these methods are unable to predict the dynamic force of tools and soil behavior because of non-linear soil properties. A simulation experiment was conducted using the DEM-MBD coupling method to explore soil disturbance caused by cavity seeders and the resistance to entry. Additionally, the effect of the maize shape and the cavity planter motion on the seed number qualification and the empty cavity rate was investigated. It was proposed that the inverted hook be used to prevent the movement of maize seeds up and down in cavity seeders, thereby improving seed filling performance. Simulations and experiments were conducted, and the results showed that the average empty cavity rate and the seed number qualification were 2.0% and 91.3%, respectively, which met the requirements of the maize sowing standards.

**Keywords:** maize; soil; cavity planter; interaction mechanism; DEM; verification

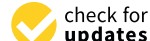



## 1. Introduction

The main working process of the cavity planter with vertical insertion is to vertically insert the hole seeder into the soil and discharge the maize seeds into the hole. Therefore, the cavity planter mainly interacts with maize seeds and soil. The cavity planter's optimization is also based on them. Soil mechanics is a priority issue when designing and optimizing components to reduce resistance and consumption in agricultural tillage, precision seeding, and efficient harvesting [1]. As soil separation, mixing, cracking, and flow belong to the characteristics of a bulk particle, the discrete element method (DEM) is the best research choice. Initially, DEM was used for the movement analysis of rock slopes [2]. Subsequently, it has been widely developed in various fields, and DEM is mainly used in agricultural engineering to study the interaction between tools and soils. DEM has proved its reliability in predicting soil resistance by means of a deep pine shovel–soil interaction model [3,4]. The shape of the soil particles and the bond between the soil particles affect the resistance of the tool. Ono et al. [5] created six different shapes of soil particles with DEM and compared the working resistance of the bulldozers. The non-cohesive/cohesive particle model and Newtonian fluid/non-Newtonian fluid wet particle of soil cultivation model were analyzed by DEM [6]. Zhang et al. [7,8] added liquid bridge forces due to moisture to the particle contact model to simulate the cohesive interaction between soil particles using DEM. The slant-shank folding shovel's structure and parameters are optimized with the aid of DEM [9]. In addition, with DEM the maize models built by the spherical aggregation method are in good agreement with the experimental results in terms of bulk density,



repose angle, and flow motion [10,11]. In this regard, it is evident that DEM research on the interaction between tools and soil is on the rise.

Meanwhile, a cavity planter's performance is affected not only by the soil and maize, but also by the interaction between the non-circular gear drive forward speed compensation mechanism and the overall speed of the machine. In practice, the interaction between the cavity planter and the soil involves the compaction of the soil by the wheels, the support of the soil on the wheels and the sliding of the soil, the disturbance of the soil caused by the cavity planters, and also the circulation of the maize seeds and seed discharge. Thus, complex interactions are involved among them. It is challenging to clearly evaluate seed motion, soil and ground wheels, soil and cavity planters, seed circulation, and seed motion. It should be noted that the forward speed compensation mechanism on the cavity planter with vertical insertion is driven by a non-cylindrical gear, and that its contact motion is complex, and thus cannot be achieved by simple motion settings in EDEM, so it must be achieved using virtual prototyping software [12]. There are clear advantages to using the Korean FunctionBay software in the areas of large model calculations, sliding and collision contact, and the design and optimization of flexible bodies in motion [13].

As DEM and multibody dynamics (MBD) have developed, the coupled DEM-MBD numerical simulation technique has gained widespread use in industry [14]. Combining DEM with MBD is an excellent approach for understanding the interaction between maize seeds, soil, and forward speed compensation mechanisms. This paper focuses on the DEM-MBD coupled simulation and the self-developed 2BZ-2 cavity planter with vertical insertion. It examines the effect of various factors on soil disturbance and cavity planter performance to propose a plan and ideas for optimizing the direct hole sowing machine.

## 2. Materials and Methods

### 2.1. Virtual Prototype Establishment of the Cavity Planter with Vertical Insertion

The maize cavity planter with vertical insertion consists of a frame, engine, traction wheel, gearbox, differential, forward speed compensation mechanism, cavity seeder, outer grooved wheel seeder, guide wheel, guide frame, and handle, as demonstrated in Figure 1. The cavity seeder is the main part of the seeder, which participates in the interaction process between corn and seeder. A core component in the cavity seeder is the shaped groove, as shown in Figure 1c. The engine provides the power for the cavity seeding process, partly driving the traction wheel, and partly driving the forward speed compensation mechanism. The gearbox transmits the power input from the engine to the differential speed and forward speed compensation mechanism after speed regulation, shunting and reversing; the differential speed can achieve the steering of the traction wheel; the forward speed compensation mechanism is the core component of the direct maize cavity on film. The forward speed compensation mechanism is the core component of the maize cavity planter with vertical insertion, and is mainly responsible for "zero speed" processing of the traction speed of the whole machine during the seeding process, i.e., it generates movements that offset the horizontal speed of the whole machine so that the horizontal direction before and after insertion is approximately stationary with respect to the ground. The outer grooved wheel seeder provides an even seed supply to the seeder so that the seed in the seed tube of the seeder always maintains a seed volume conducive to seeding. The guide wheel and guide frame facilitate the adjustment of the seeder, and the handle facilitates manual operation. The angle between the crank rods of the two sets of forward speed compensation mechanisms is set at 180°, which improves the smoothness of the machine.

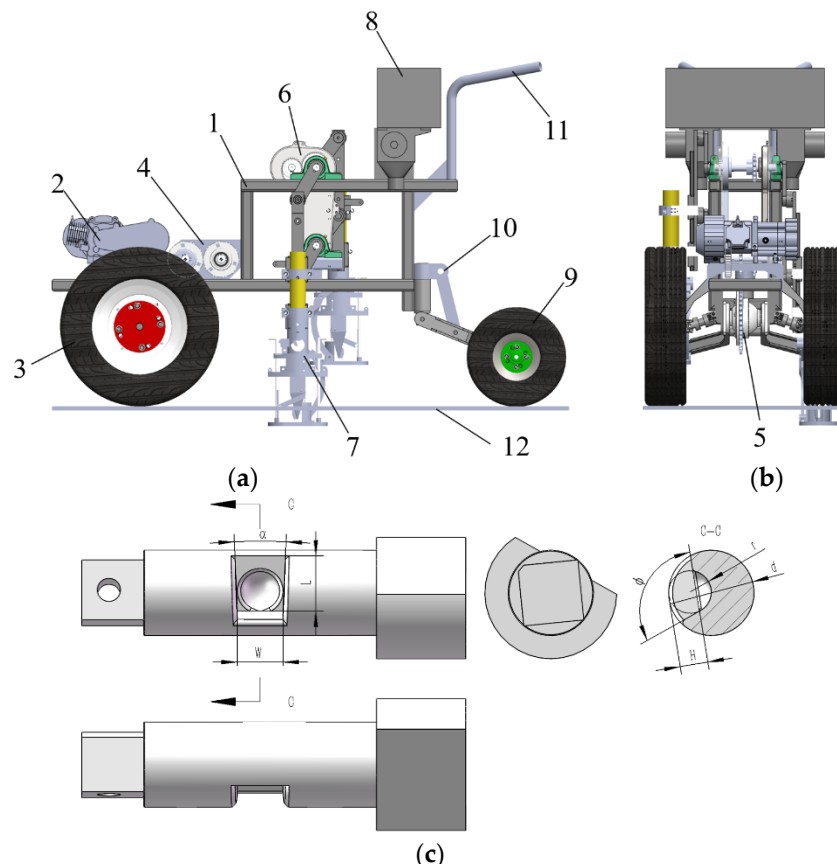

**Figure 1.** Structural components of the maize cavity planter with vertical insertion. (**a**) Main view. (**b**) Front view. (**c**) Structure schematic of shaped groove. (1) frame; (2) engine; (3) traction wheel; (4) gearbox; (5) differential; (6) forward speed compensation mechanism; (7) cavity seeder; (8) outer grooved wheel seed discharger; (9) guide wheel; (10) guide frame; (11) handle; (12) ground. Note, $L$ is the length of shaped groove. $W$ is its width. $H$ is the height. $\alpha$ is opening angle of the groove. $d$ is the diameter of seed taking wheel. $r$ is the maximum diameter of the spherical hole. $\varphi$ is the included angle of the inner groove line.

There is a non-circular gear that drives a complex drive system, which is assembled from virtual prototypes, to compensate for the forward speed of the cavity planter with vertical insertion. Virtual prototypes are constructed using constraints, motion, and drive information. In MBD simulation process, the parts that do not affect the simulation results can be omitted or simplified, thereby reducing the number of unnecessary constraints and parts. The specific operation is to export the 3D model x_t format, and import to Recurdyn V9R2 (FunctionBay Co., Seongnam, Korea), delete the redundant parts such as bolts and bearings, and delete the parts including bearings, mounting screws, pins, drive chains, gears, etc. Establish rotation, translation, and fixation constraints between components, and set related motion, add motion information based on rotation and translation subs. As shown in Figure 2b, the virtual prototype model consists of engine power output, speed change and reversing, ground drive system, transmission ratio conversion system, direct cavity seeder system, quantitative seed discharge system, and passive guide wheel system. Ground drive system includes differential mechanism. The cavity planter can turn flexibly on the ground. The traction wheel can be turned at different speeds by turning the handle.

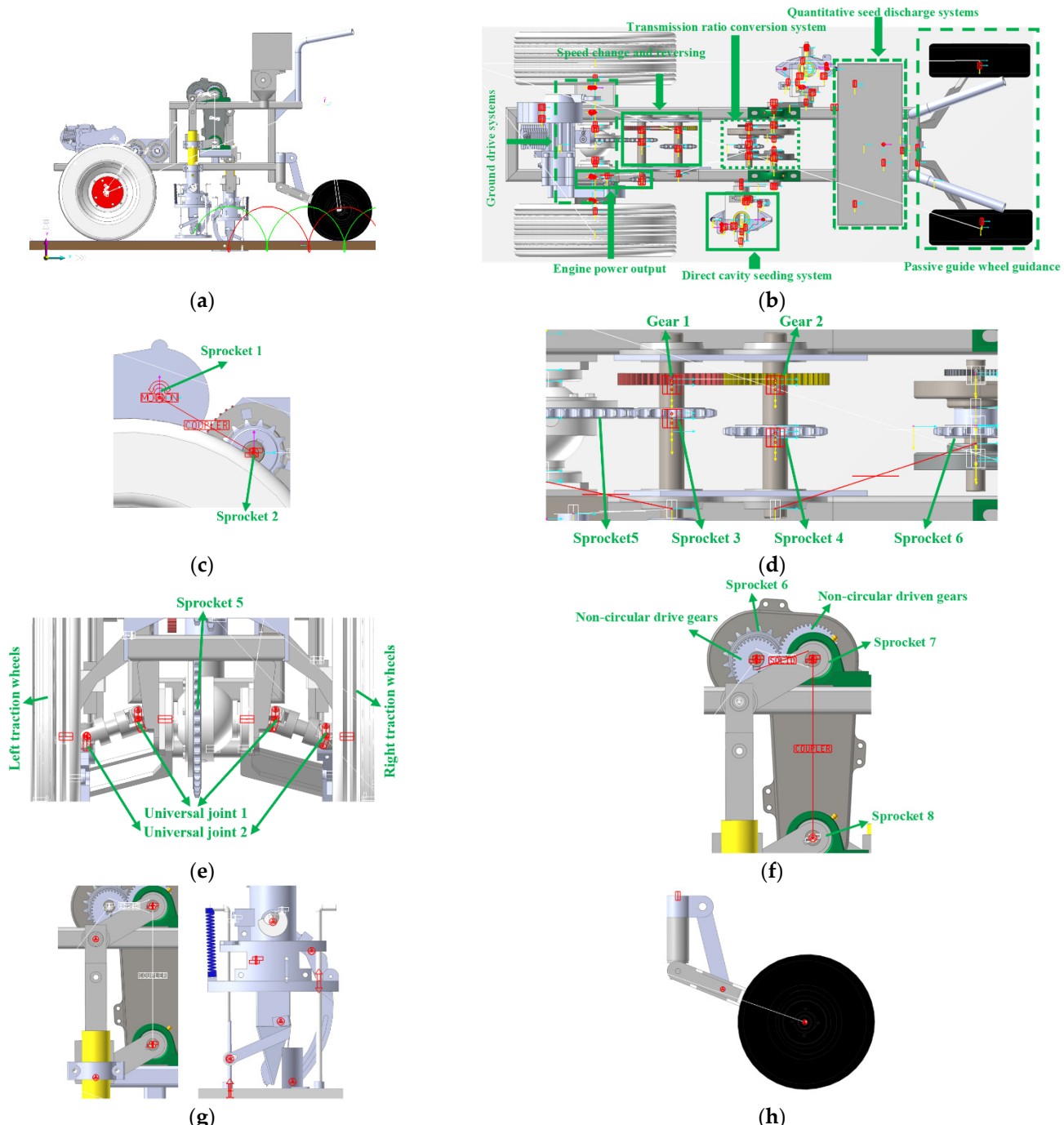

**Figure 2.** Constraint relationships of the cavity planter with vertical insertion. (**a**) Virtual prototype model. (**b**) Drive train division. (**c**) Engine power output. (**d**) Speed changing and reversing. (**e**) Ground drive system. (**f**) Transmission ratio conversion system. (**g**) Cavity seeding system. (**h**) Passive wheel guidance.

The working process is specifically that sprocket 1 is driven by engine power output to turn sprocket 2, and sprocket 2 is connected to sprocket 1 by a couple, and the transmission ratio is 1:1 in Figure 2c. The variable speed reversing power input shaft is hinged to the frame and is fixed to the power input sprocket. After speed changing and reversing, the power is distributed to the ground drive system and transmission ratio conversion system. Sprocket 3 in the speed changing and reversing is connected to sprocket 5 from the ground drive system, and the transmission ratio is 1:3.2, while gear 1 engages with the passive gear

2 that is hinged to the frame and fixedly connected to sprocket 4, as seen in Figure 2d. Then it is connected to sprocket 6 with chain. It drives the active non-circular gear to bring driven non-circular gears with variable ratios. It is fixedly connected to sprocket 7, then that drives sprocket 8 and the transmission ratio is 1:1, enabling cavity seeding with vertical insertion, as shown in Figure 2f. In another method, sprocket 5 that is mounted on the ground drive system drives universal joint 1 and universal joint 2 to make traction wheels rotate, as indicated. Other constraint relations are presented in Figure 2e,f for cavity seeding system and passive wheel guidance. The completed virtual prototype model is shown in Figure 2a.

## 2.2. Soil Modeling

Based on the soil type of China's northwest dry zone, a simplified soil simulation model was developed to improve simulation efficiency. Specifically, soil model (length 3100 mm × width 1000 mm × height 320 mm) was created using SolidWorks 2018.sp5, saved as. x_t format and then imported into EDEM 2018 (Altair co., Richardson, TX, USA) through the geometry option [15–17]. Hysteretic Spring and Linear Cohesion were selected to simulate sandy loam, and the related simulation parameters are shown in Table 1. The Hertz–Mindlin model and Linear cohesion model were chosen to simulate sandy soils and the simulation parameters are seen in Table 2. The soil particles' radius was set to 10 mm, and the variance of the radius was 0.05. The soil simulation parameters are shown in Table 3 at the same level. Therefore, the simplified model is indicated in Figure 3.

**Table 1.** Predicted simulation parameters of sandy loam.

| Water Content % | Coefficient of Static Friction | Yield Strength/kPa | Rolling Friction Coefficient | Stiffness Factor | Damping Factor | Coefficient of Restitution | Cohesion Strength/kPa |
|---|---|---|---|---|---|---|---|
| 1 | 0.06 | $10.38 \times 10^6$ | 0.01 | 0.73 | 0.95 | 0.6 | 7.04 |
| 6 | 0.05 | $9.55 \times 10^6$ | 0.01 | 0.73 | 0.95 | 0.6 | 6.13 |
| 12 | 0.05 | $8.89 \times 10^6$ | 0.01 | 0.73 | 0.95 | 0.6 | 6.14 |
| 18 | 0.03 | $8.78 \times 10^6$ | 0.01 | 0.73 | 0.95 | 0.6 | 3.64 |

**Table 2.** Predicted simulation parameters of sandy soil.

| Water Content % | Coefficient of Static Friction | Rolling Friction Coefficient | Coefficient of Restitution | Cohesion Strength/kPa |
|---|---|---|---|---|
| 1 | 0.1 | 0.72 | 0.6 | 11.04 |
| 6 | 0.07 | 0.68 | 0.6 | 8.09 |
| 12 | 0.02 | 0.66 | 0.6 | 3.26 |
| 18 | 0.02 | 0.59 | 0.6 | 3.14 |

**Table 3.** Simulation parameters.

| Parameter | Value | | | | Source |
|---|---|---|---|---|---|
| Water content of sandy soil/sandy loam/% | 1 | 6 | 12 | 18 | |
| Soil particle density of sandy soil/sandy loam/(kg/m$^3$) | 1600/1200 | | | | Measurement |
| Poisson's ratio of sandy soil/sandy loam | 0.3/0.4 | | | | Literature [18,19] |
| Shear modulus of sandy soil/sandy loam/MPa | 11.5/3.27 | | | | Calculation [18,19] |
| Steel density/(kg/m$^3$) | 7850 | | | | Literature [20] |
| Steel Poisson's ratio | 0.3 | | | | Literature [20] |
| Steel shear modulus/MPa | $7.9 \times 10^4$ | | | | Literature [20] |

**Table 3.** *Cont.*

| Parameter | Value | Source |
|---|---|---|
| Coefficient of static friction between soil and steel | 0.5 | Literature [20] |
| Coefficient of rolling friction between soil and steel | 0.05 | Literature [20] |
| Coefficient of restitution between soil and steel | 0.6 | Literature [20] |
| Rubber density/(kg/m$^3$) | 940 | Literature [16] |
| Rubber Poisson's ratio | 0.47 | Literature [16] |
| Shear modulus of rubber/MPa | $2.9 \times 10^3$ | Literature [16] |
| Coefficient of static friction between soil and rubber | 0.57 | Literature [16] |
| Coefficient of rolling friction between soil and rubber | 0.31 | Literature [16] |
| Coefficient of restitution between soil and rubber | 0.6 | Literature [16] |

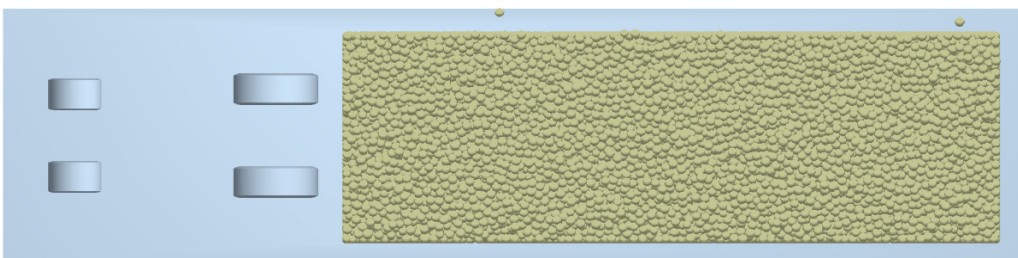

**Figure 3.** Soil simulation simplified model.

*2.3. Maize Seed Modeling*

Intercropping effects with maize also occur with the cavity seeder. The effect of maize shape on the seeding performance was investigated. Three typical shapes of maize models were used. Among them, the horse tooth seed consisted of nine unequal spherical particles, the spherical cone consisted of three unequal radius spherical particles, and the spherical particle consisted of one spherical particle, as presented in Figure 4.

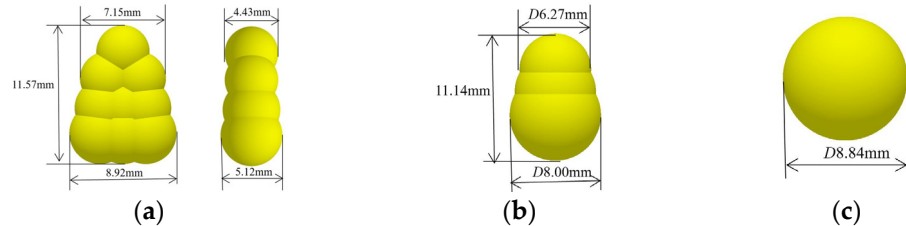

(**a**)         (**b**)         (**c**)

**Figure 4.** The discrete element model of maize seeds. (**a**) Horse tooth. (**b**) Spherical cone. (**c**) Spherical.

Maize simulation parameters were set as: Poisson's maize model ratio was 0.4, the shear modulus was $1.37 \times 10^8$ Pa, and the density of maize grains was 1200 kg/m$^3$. Interaction parameters between maize seeds included the coefficient of restitution, the coefficient of static friction, and the coefficient of rolling friction. The coefficient of restitution was set to 0.37, and the coefficient of static friction was set to 0.2. The coefficients of rolling friction of horse tooth, spherical cone, and spherical maize seeds were 0.013, 0.024, and 0.053, respectively. The coefficients between horse tooth and spherical cone, horse tooth and spherical, spherical cone and spherical maize models were 0.004, 0.003, 0.014, respectively. The coefficient of restitution between maize and steel was set to 0.37, the coefficient of static friction was set to 0.408, and the coefficient of rolling friction was 0.01 [21].

### 2.4. Coupling Modeling

During the coupling simulation of direct cavity sowing with soil and maize interactions, the parts of the virtual prototype that interact with the discrete element method soil and maize models need to be output in the form of walls. In the process of coupling simulation jointly with EDEM and RecurDyn, RecurDyn receives the external force from EDEM through the wall, and EDEM software receives the movement data from the contact geometry from RecurDyn through the wall. Based on the two-way DEM-MBD coupling method, maize seeds' complex motion and force process can be observed visually using discrete element software [22]. During the RecurDyn coupling setup, files were created and exported for the parts in contact with the particles, including two traction wheels, two guide wheels, two fixed and two movable cavity formers, and two support plates. The wall files were imported using the Import Geometry from RecurDyn function under the geometry's module in EDEM. The soil contact parts of the cavity planter include the traction wheel, the guide wheel, the fixed cavity seeder, the movable cavity seeder, and the support plate, with the traction wheel and guide wheel made of rubber and the fixed cavity seeder, movable cavity seeder, and support plate made of steel. The simulation model of the direct cavity seeder and soil interaction is shown in Figure 5.

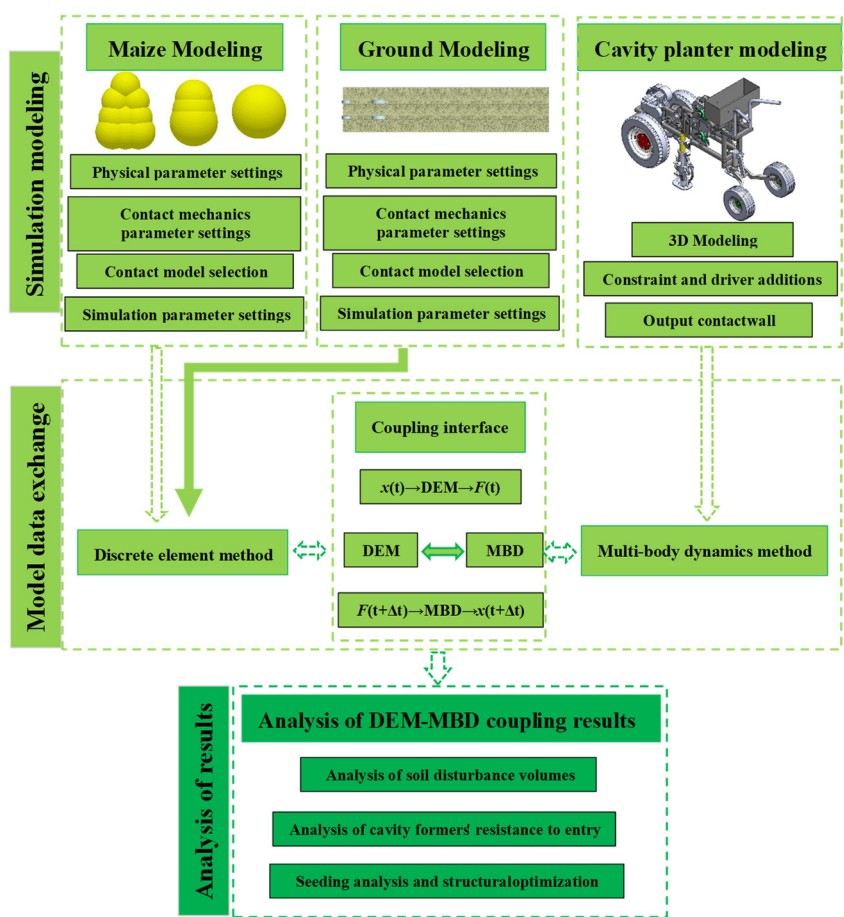

**Figure 5.** The flow on DEM-MBD coupling simulation.

## 3. Results

### 3.1. Coupling Process

It is necessary to conduct separate studies on the mechanism of intercropping between maize seeds, soil, and the cavity seeding considering the inconsistency of the particle contact models used for soils and maize seeds. Marker points were established at the two cavity seeder tips to track their motion trajectory. By measuring the course of the cavity seeder during travel, a transition section of 1200 mm was placed between the contact of the

cavity seeder and the soil model. The transition section allowed comparison of the effect of the ground wheel slip on the trajectory. Figure 5 illustrates the coupling process between the cavity seeder and soil in a simulation.

Figure 6 shows that the cavity seeder travels smoothly on the soil model, that the forward speed compensation mechanism is usually engaged, and that the seeder enters the soil at a suitable depth. From the trajectory, it is evident that the forward speed compensation mechanism has a good compensation effect in the transition section, and a weaker compensation effect in the coupling section. Research was conducted based on the interaction model to determine the influence of soil type and water content on soil disturbance and entry resistance. Several factors affect the impact of forward speed compensation, including soil conditions, traction wheels, and the input speed ratio of the forward speed compensation mechanism. During field operations, soil conditions and speed ratio can be selected at the right time.

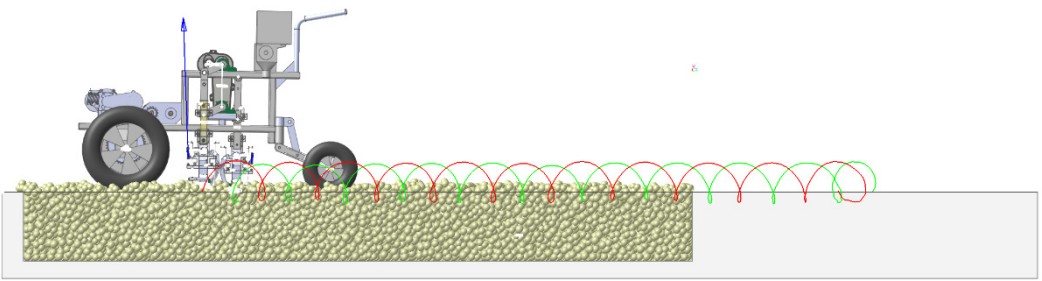

**Figure 6.** Simulation process of cavity seeding.

### 3.2. Effect of Soil Type on Soil Disturbance

There is a difference in compressibility between the sandy soils and sandy loams studied in this paper, but they both possess a certain degree of cohesion. For sandy soils, the Hertz-Mindlin (no slip) and Linear Cohesion models were selected, and for sandy loam soils, the Hysteretic Spring and Linear Cohesion models were selected. The cavity seeder slips and sags as it travels on the soil, which affects its forward speed compensation. Figure 7 provides a comparison of the forward speed of the cavity seeder for the two soil types at 1% water content.

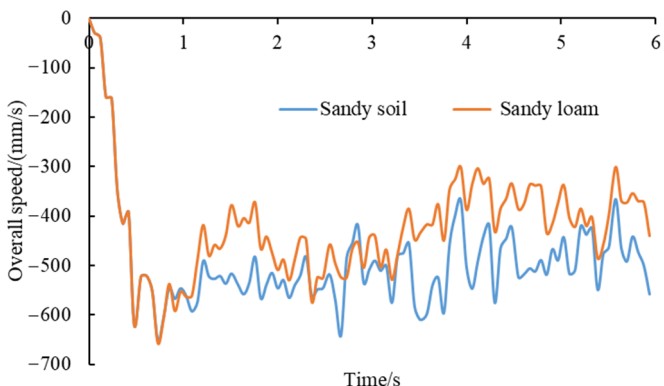

**Figure 7.** Overall speed of the cavity planter travelling on two different soils.

Figure 7 illustrates how soil type affects forward speed. In sandy loam, the average speed is −487.757 mm/s, and in sandy soil, the average speed is −420.035 mm/s. The cavity seeder produces less slip in sandy soil than in sandy loam because of the lower speed in sandy loam. It is necessary to compare the amount of soil drop between the two soil types since soil drop probably causes the seeder's speed to decrease, as shown in Figure 8.

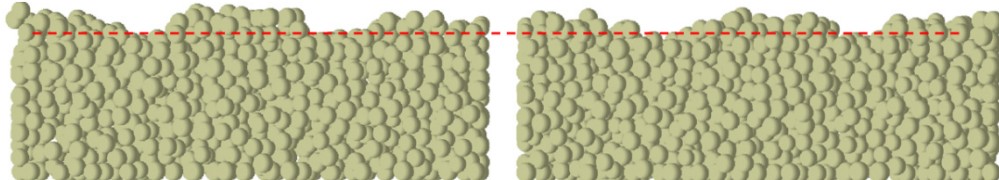

**Figure 8.** Amount of soil drop for two types of soil.

In both sandy soil and sandy loam models, the amount of soil drop is essentially similar, as seen in Figure 8. Accordingly, the cavity seeder slips on the soil model, and the amount of soil drop is not the primary cause.

### 3.3. Effect of Water Content on the Amount of Soil Disturbance

The influence of the two types of soils on the trajectory of the cavity seeder under different water content conditions was analyzed separately. The left and right cavity seeders were compared, as illustrated in Figure 9.

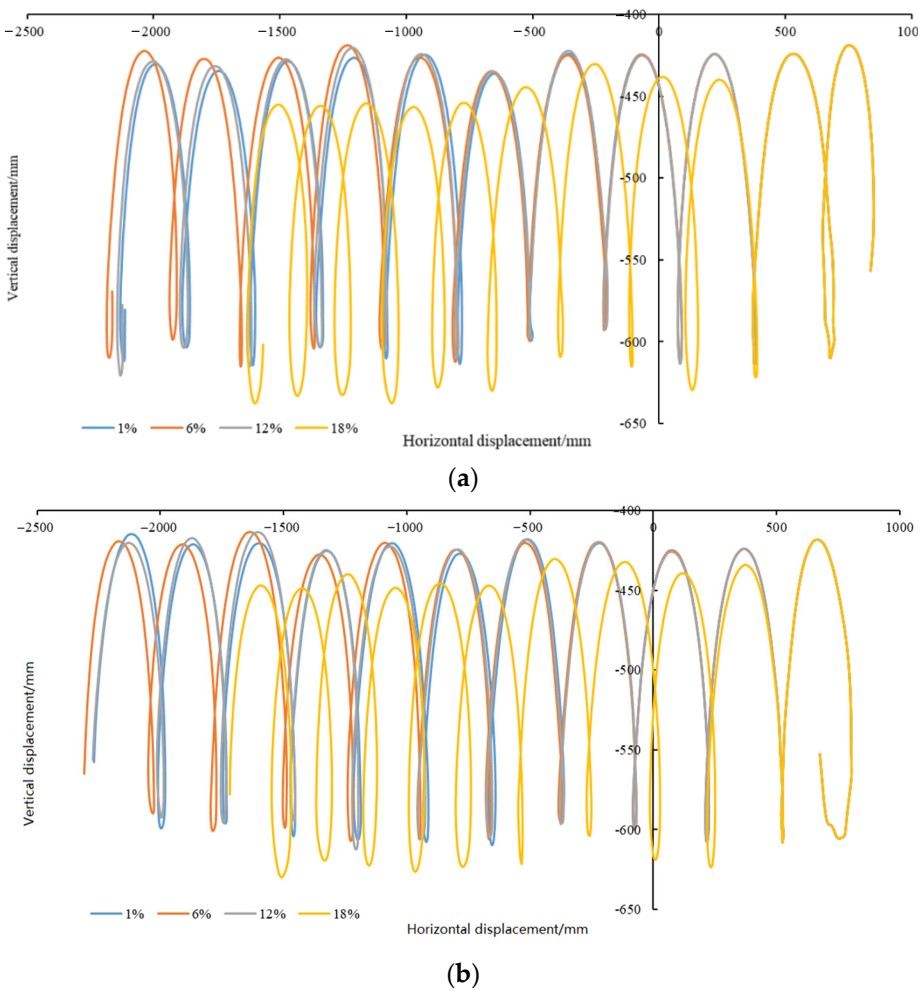

**Figure 9.** Trajectory of cavity seeder's movement under sandy soil. (**a**) Right cavity seeder. (**b**) Left cavity seeder.

Figure 9 shows that sandy soil with varying water content affects the trajectory of the cavity seeder. The greatest distance was at 6%, the nearest at 18%, and the closest at 1% and 12%. According to the displacement in the vertical direction, there was a small amount of soil drop when the water content was between 1 and 12%. Comparatively, that was evident when the water content was 18%. Does the cavity seeder pattern in sandy loam

correspond to that in sandy soil? Figure 10 illustrates the trajectory of the cavity seeder under sandy loam.

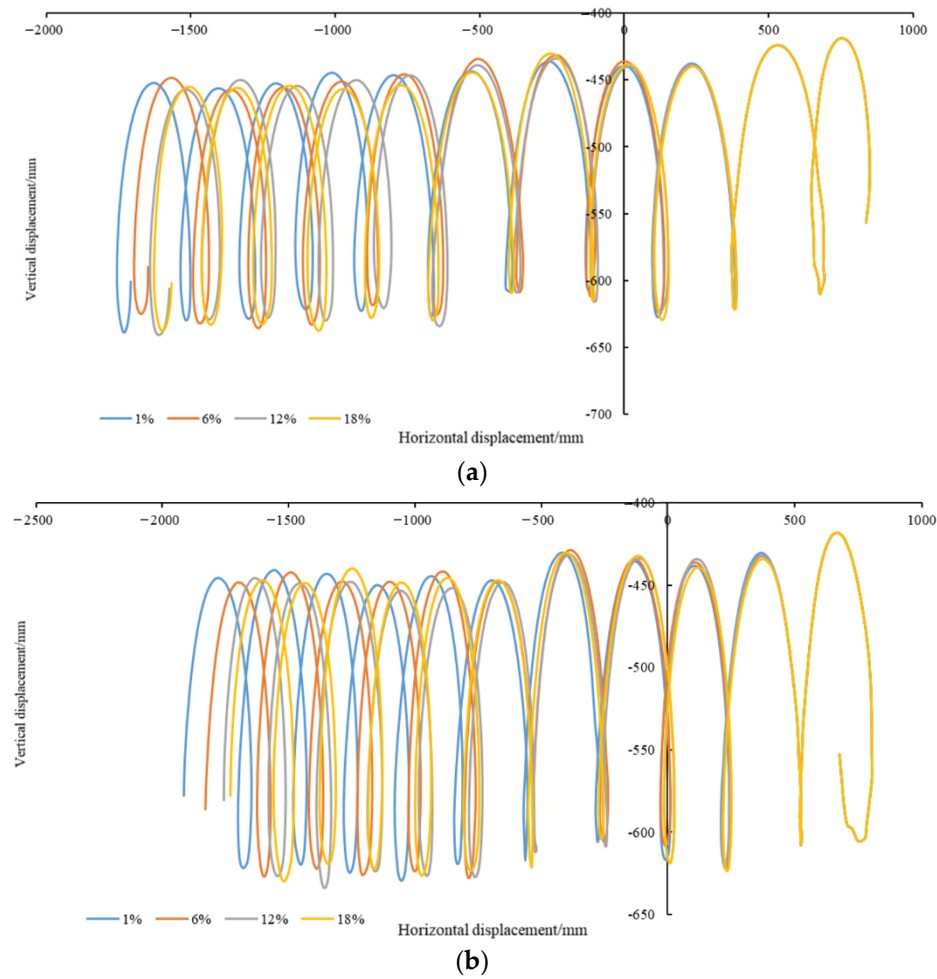

**Figure 10.** Trajectory of cavity seeders under sandy loam. (**a**) Right cavity seeder. (**b**) Left cavity seeder.

The trajectory of the cavity seeders was affected by sandy loam soil with different water contents, as shown in Figure 10a. The trajectory decreased from 1% to 18%, indicating that the slip rate of the cavity seeder increased as the water content increased. In terms of displacement in the vertical direction, the cavity seeder dropped more when the water content ranged from 6 to 18% as compared to 1%. Under all three conditions of water content, the amount of soil drop was equal. Figure 10b provides further evidence.

According to Figures 9 and 10, the slip rate of the cavity seeder traveling on sandy loam was significantly higher than that of sandy soil under the same water content conditions. Sandy loam soils performed 19% better than sandy soils when the water content was 1%. In sandy loam, the drop was 24% higher than in sandy soil when the water content was 6%. Sandy loam slips were 26% better than sandy soil when the water content was 12%. Sandy loam and sandy soil had similar slip amounts when the water content was 18%. It appears from the above results that the slip of sandy loam increases more rapidly with increasing water content than that of sandy soil. Nevertheless, both soil types were comparable when the water content was 18%. It seems that the input speed ratio between the traction wheel and the forward speed compensation mechanism needs to be adjusted for different types of soil to reduce soil disturbance. During cavity seeding, the soil water content should not exceed 18%. Next, we investigate the influence of the input speed ratio between the traction wheel and the forward speed compensation mechanism on soil disturbance.

### 3.4. Influence of the Rotation Speed Ratio on the Amount of Soil Disturbance

An investigation was conducted to determine the effect of traction wheel speed on soil disturbance. According to the theoretical analysis, an increase in the traction wheel speed will result in an increase in the forward speed of the cavity seeder. However, the forward speed compensation mechanism will remain unchanged. Five speed ratios are listed in Table 4. As a result, the cavity seeder will move backward relative to the ground. Cavity seeders move slowly relative to the ground when the speed of the traction wheels decreases. Large displacements forward and backwards can disrupt the soil disturbance and change the seed germination environment in the cavity hole. Figure 11 illustrates the trajectory of cavity seeding under different transmission ratio conditions.

**Table 4.** Rotation speed ratio.

| Parameters | Value | | | | |
|---|---|---|---|---|---|
| Rotation speed of forward speed compensation mechanism/(rad/s) | 11.50 | 11.50 | 11.50 | 11.50 | 11.50 |
| Rotation speed of traction wheel/(rad/s) | 4.44 | 3.72 | 3.2 | 2.81 | 2.51 |
| Ratio | 2.59 | 3.09 | 3.59 | 4.09 | 4.59 |

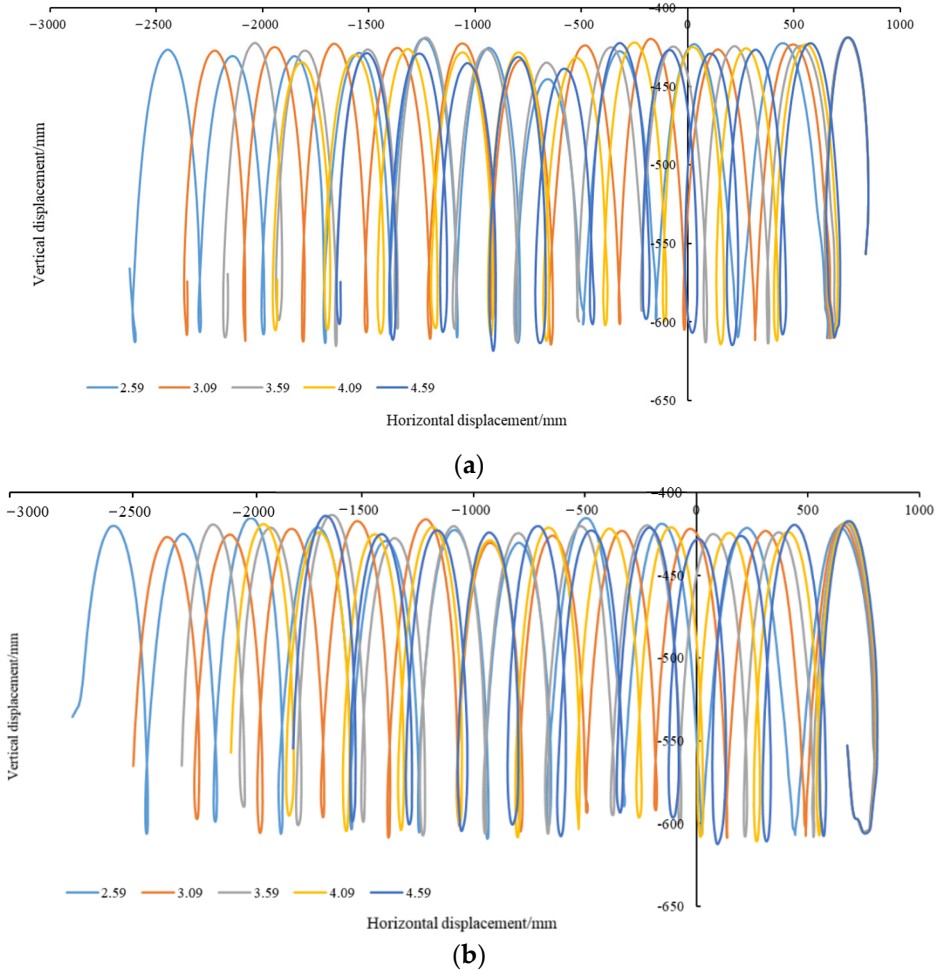

(a)

(b)

**Figure 11.** Trajectory of cavity seeders under different rotation speed ratios. (**a**) Right cavity seeder. (**b**) Left cavity seeder.

According to Figure 11, as the speed of the traction wheel decreases, the trajectory of the cavity seeder gradually moves backward. In the forwarding speed compensation mechanism, the horizontal speed cannot be compensated, so the intersection of trajectory shifts upward, and the theoretical seeding depth increases. As the ratio decreases, the traction wheel speed increases, and the motion trajectory gradually advances. There is no doubt that the cavity seeder will cause soil disturbance due to the different types of soil. Due to the deeper soil disturbance caused by the cavity seeder, the seed growth layer will be destroyed. Furthermore, the cavity seeder can loosen the soil above the seed and allow air to enter the seed bed. As a result, it is recommended to moderately disturb the soil above the intersection of trajectory to enlarge the hole seeder and increase air circulation.

### 3.5. Effect of Water Content on the Resistance to Soil Entry

The resistance of the cavity seeder to soil affects the power input of the cavity seeder and the smoothness of the machine [23]. It is necessary to examine the effect of two soil types with four water contents on the resistance of the cavity seeder. The purpose of this is to provide some support for the cavity seeder. Figure 12 illustrates the resistance of the cavity seeder.

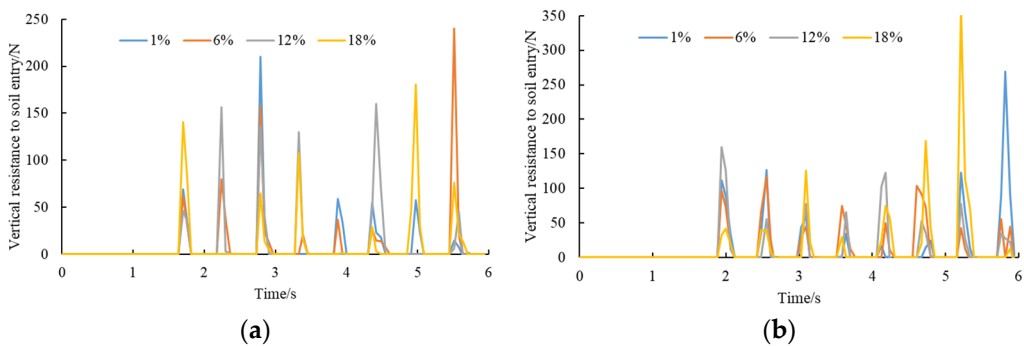

**Figure 12.** Entry resistance of the cavity seeder in sandy soil. (**a**) Left cavity seeder. (**b**) Right cavity seeder.

As shown in Figure 12a, the water content of sandy soil also affects the entry resistance of cavity seeders. The maximum resistance for the four water contents can be compared based on their average values. As a result, the average value was 57.99 N at 1%, 78.09 N at 6%, 84.29 N at 12%, and 78.03 N at 18%. It was found that when the water content was 1–12%, the entry resistance increased on the left side and decreased when the water content was 18%. According to Figure 12b, the average value was 88.55 N at 1%, 72.97 N at 6%, 76.39 N at 12%, and 106.90 N at 18%. It is evident that the entry resistance increased when the water content ranged from 6 to 18%. Figure 13 illustrates the resistance to cavity penetration in sandy loam soils.

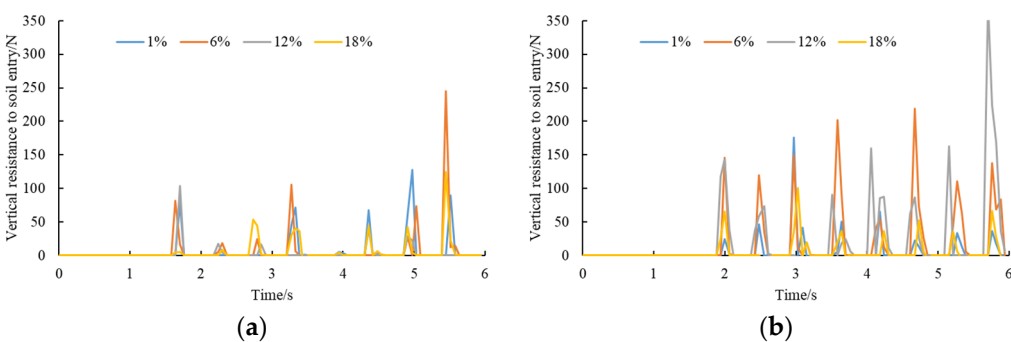

**Figure 13.** Entry resistance of the cavity seeder in sandy loam. (**a**) Left cavity seeder. (**b**) Right cavity seeder.

According to Figure 13a, the water content of sandy loam soils also influences entry resistance. For each of the four water content conditions, the average value of the maximum penetration resistance of sandy loam soil can be compared. According to the results, the average value was 55 N at 1%, 78.42 N at 6%, 29.35 N at 12%, and 40.52 N at 18%. There was an increase in entry resistance when the water content was between 1 and 6%. In contrast, there was a decrease in entry resistance when the water content was between 6 and 12%. Figure 13b shows average values of 56.83 N at 1%, 142.35 N at 6%, 127.78 N at 12%, and 49.18 N at 18%. The entry resistance increased when the water content was between 1 and 6%. In the range of 6–18% water content, the entry resistance decreased. As the moisture content of the sandy loam soil increased, the resistance first increased and then decreased. Compared to sandy soil, sandy loam has a significantly higher cavity resistance.

During the insertion of the cavity seeder into the soil, the duckbill is subjected to resistance from the soil. At the same time, the duckbill is also strongly stressed against the soil. A cloud plot of the change in stress between the duckbill and the soil from the start insertion to near the bottom of the cavity is observed in a sandy loam soil with a moisture content of 6%.

From Figure 14a, the fixed duckbill stress remains blue, but the stress is changing, transitioning from $1.11 \times 10^5$ Pa to $1.35 \times 10^5$ Pa. From Figure 14b, the axial stress of the duckbill on the soil appears red, with the soil stress at $1.39 \times 10^5$ Pa when the duckbill first enters the soil, gradually decreasing to $7.96 \times 10^4$ Pa before increasing in small floats to $9.10 \times 10^4$ Pa.

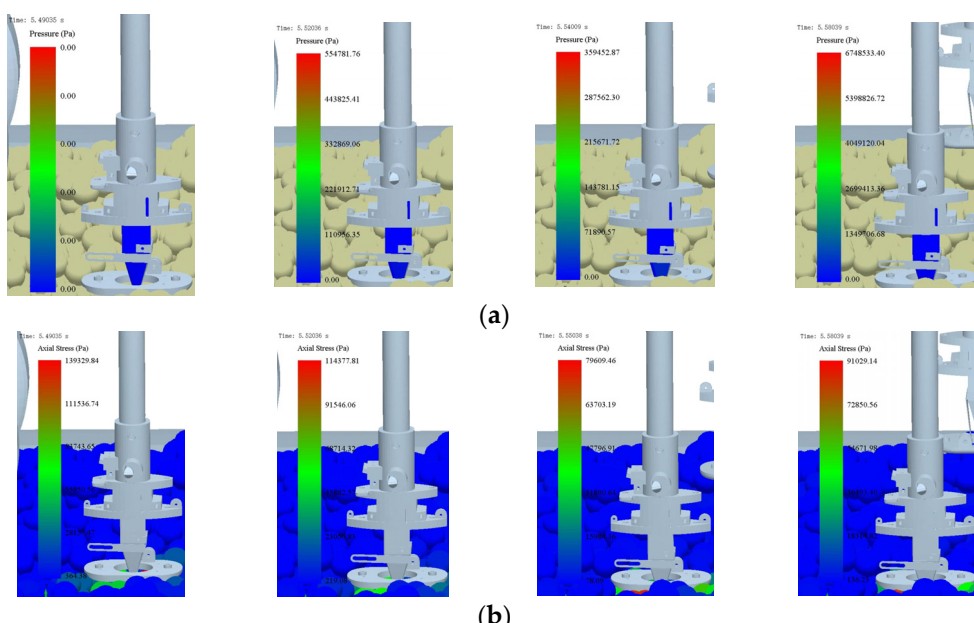

**Figure 14.** Entry soil stress change in the cavity seeder in sandy loam. (**a**) Duckbill stress. (**b**) Soil stress.

### 3.6. Effect of Maize Type on Seeding Performance

In addition to seeding performance, seeding discharge is affected by the movement of the direct hole sowing process. Among these, maize seed filling nest performance is the most significant [24,25]. The key to improving seeding quality is to improve seed filling performance [26]. There are several factors that influence the seed filling process, such as the maize shape, the shape of the hole, the rotation speed of the wheel that picks up seeds, and the motion of the cavity seeder. In particular, the maize shape and the movement of the cavity seeder have a greater influence on seed filling. As a first step, we examine the effect of seed shape on seeding performance. Figure 15 shows the trajectory of 50 maize seeds during seeding.

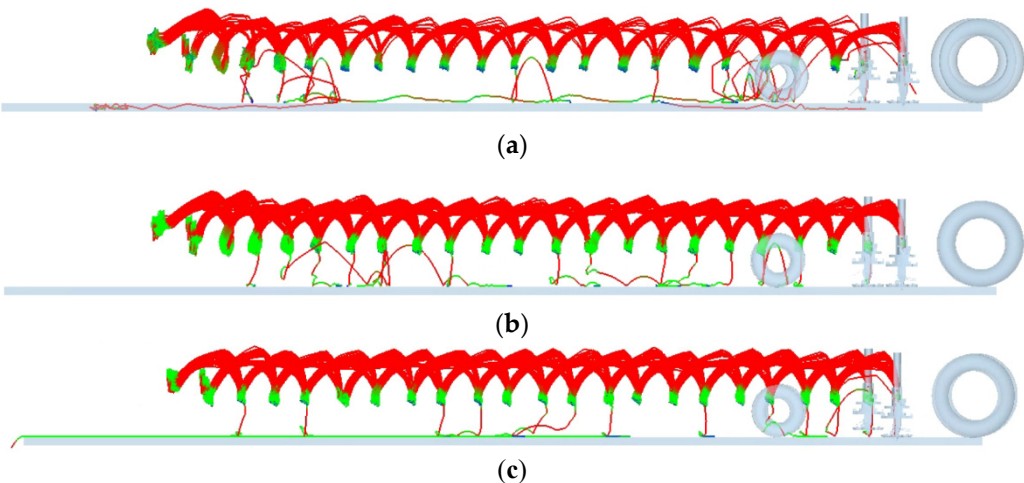

**Figure 15.** Effect of the maize shape on seeding performance. (**a**) Horse tooth. (**b**) Spherical cone. (**c**) Spherical. Note: the color of the maize seeds represents the movement speed, with red representing high speed, blue representing low speed, and green representing medium speed.

Figure 15 shows that the cavity seeder performed 22 seeding operations during the seeding process, excluding two seeding operations before the cavity seeder was stabilized. A total of 20 holes were seeded. As can be seen in Figure 15a, when sowing horse tooth shape maize, the number of qualified seeds was ten times for one seed and two times for two seeds. As a result, only 50% of seeds qualified for seeding. From Figure 15b, when seeding spherical cone shape maize, the number of seeds was 15 for one seed, and two for two seeds. Thus, 75% of seeds qualified for seeding. Figure 15c illustrates that when seeding spherical maize, 13 seeds qualified for one seed and zero for two seeds. The result was a seeding qualification rate of 65%. Figure 15 shows that the maize seed moves violently in the seed tube during filling, resulting in difficulty filling the seed. As a result, conventional hole seeders are of low quality. Additionally, the horse tooth shape maize seeds are not as well circulated as spherical cone and spherical seeds, resulting in poor seeding performance. It is necessary to mix the spherical seed with good fluidity into the horse tooth and spherical cone seed population with poor fluidity to improve the seed filling performance. Furthermore, it was found that the movement of the cavity seeder significantly affected seed filling. It is essential to improve and optimize the structure of the cavity seeder to avoid maize seeds accelerating in an upward and downward movement. Next, the motion of the horse tooth maize seeds are examined again in the cavity seeder.

*3.7. Effect of Maize Seeds' Movement in Cavity Seeder on Seed Filling Performance*

In response to the self-weight and forward speed compensation mechanism, the seed in the seed tube moves, as shown in Figure 16. Seeds remain in contact with the seed tube for too long during the filling process, resulting in a delay in the seed filling process. The maize has only 0.07 s to fill the shaped space during a cycle of 1.50 s to 1.57 s. Figure 16 illustrates the movement of horse tooth maize seeds inside a traditional structural seed tube.

From Figure 16, the maize seeds were primarily in contact with the inner wall of the seed tube during the entire process of filling the shaped slot. Even though the direction of movement of the maize seeds gradually changed from the vertical inner wall of the seed tube to the lower right side of the seed tube, the contact height between the maize and the inner wall was large, which did not facilitate seed filling. It was decided to add the inverted hook to the seed tube to separate the seed population while the seed above could slide into the inverted hook, as seen in Figure 17. A cavity seeder with the inverted hook was simulated by conducting a seed discharge simulation under the same condition. In addition, we observed the seed transport within the seed tube during 1.50~1.57 s. Figure 18 illustrates the movement of maize in the seed tube after the inverted hook was added.

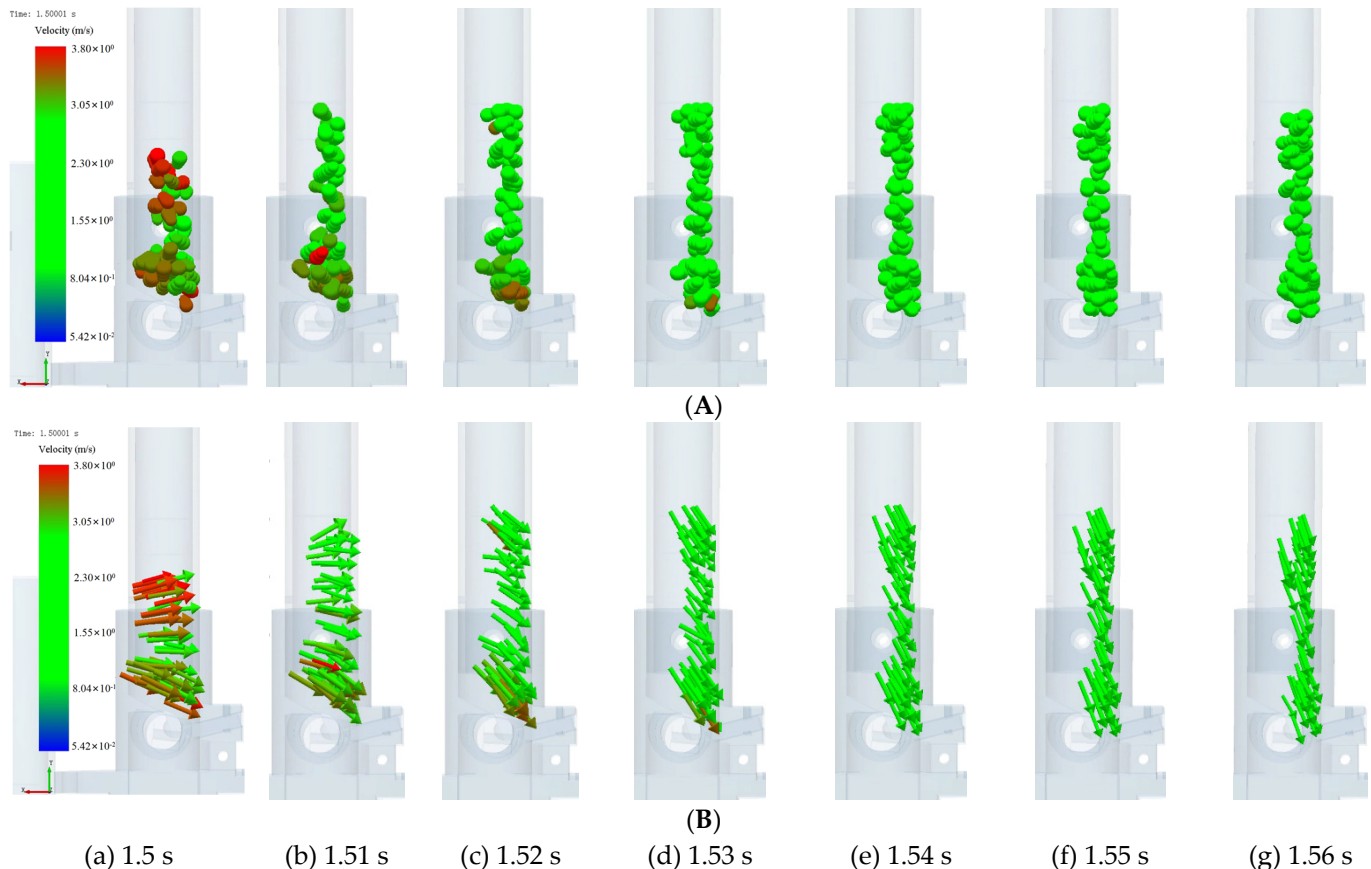

(a) 1.5 s    (b) 1.51 s    (c) 1.52 s    (d) 1.53 s    (e) 1.54 s    (f) 1.55 s    (g) 1.56 s

**Figure 16.** Movement process of horse tooth maize in traditional structural seed tube. (**A**) Default display. (**B**) Vector display.

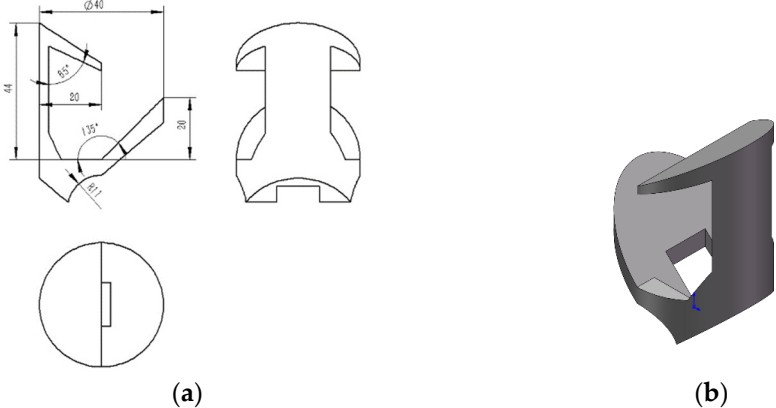

(**a**)        (**b**)

**Figure 17.** Shape and size of the inverted hook. (**a**) Two-dimensional view. (**b**) Three-dimensional view.

In Figure 18, we can see that the maize group was divided into upper and lower parts. Inverted hooks prevented maize seeds from flowing upward and backward. While seeds closer to the seed picking wheel did not move upwards, maize above could slide into the inverted curve. Nevertheless, the inverted hook did not effectively channel the slow-moving maize into the shaped groove of the seed picking wheel. Additionally, the left side structure of the inverted hook failed to function. A redesigned inverted hook for guiding seeds is illustrated in Figure 19, and the movement of the maize during seed filling in the seed tube with the inverted hook added for guiding seeds is illustrated in Figure 20.

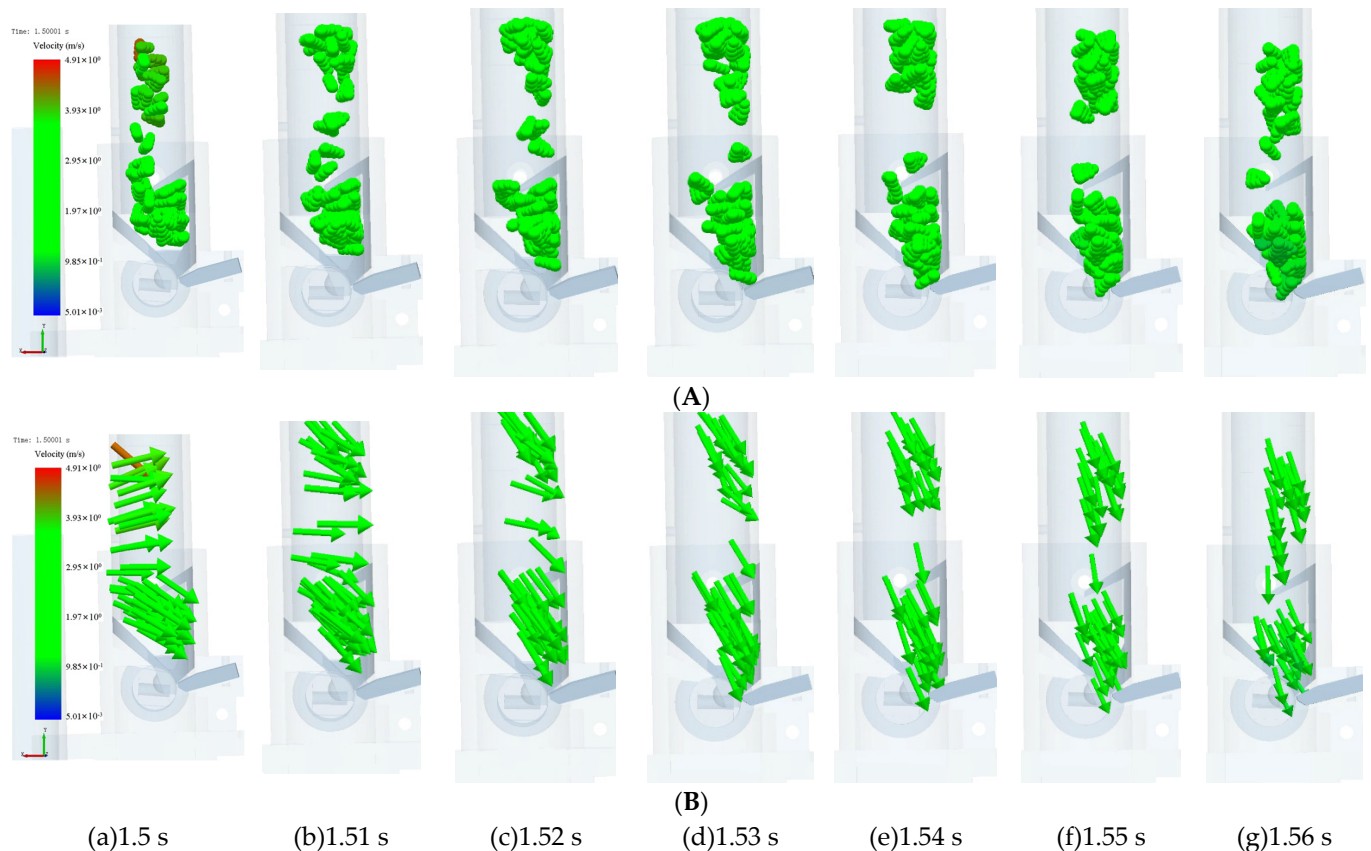

(a)1.5 s    (b)1.51 s    (c)1.52 s    (d)1.53 s    (e)1.54 s    (f)1.55 s    (g)1.56 s

**Figure 18.** Movement process of maize in the seed tube with the added inverted hook. (**A**) Default display. (**B**) Vector display.

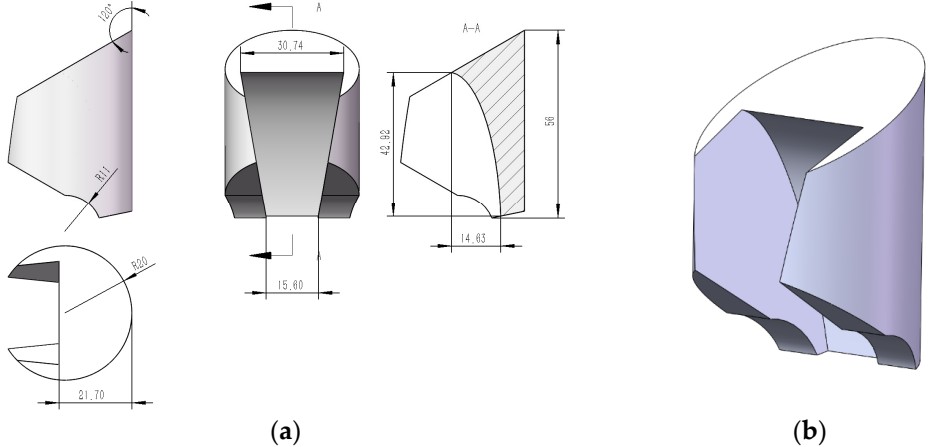

(**a**)    (**b**)

**Figure 19.** Shape and size of the inverted hook for guiding seeds. (**a**) Two-dimensional view. (**b**) Three-dimensional view.

A barbed hook is used to separate the maize in the seed tube into upper and lower populations. This is shown in Figure 20. The maize model above the inverted hook moved at a faster speed and had an uneven distribution. The maize below, on the other hand, moved uniformly and moved downward under the influence of the seed guide curve. Additionally, the guiding seed's cross-section was funnel-shaped, which enhanced the seed's filling speed. A few maize seeds can fit into the shaped groove of the seed picking wheel.

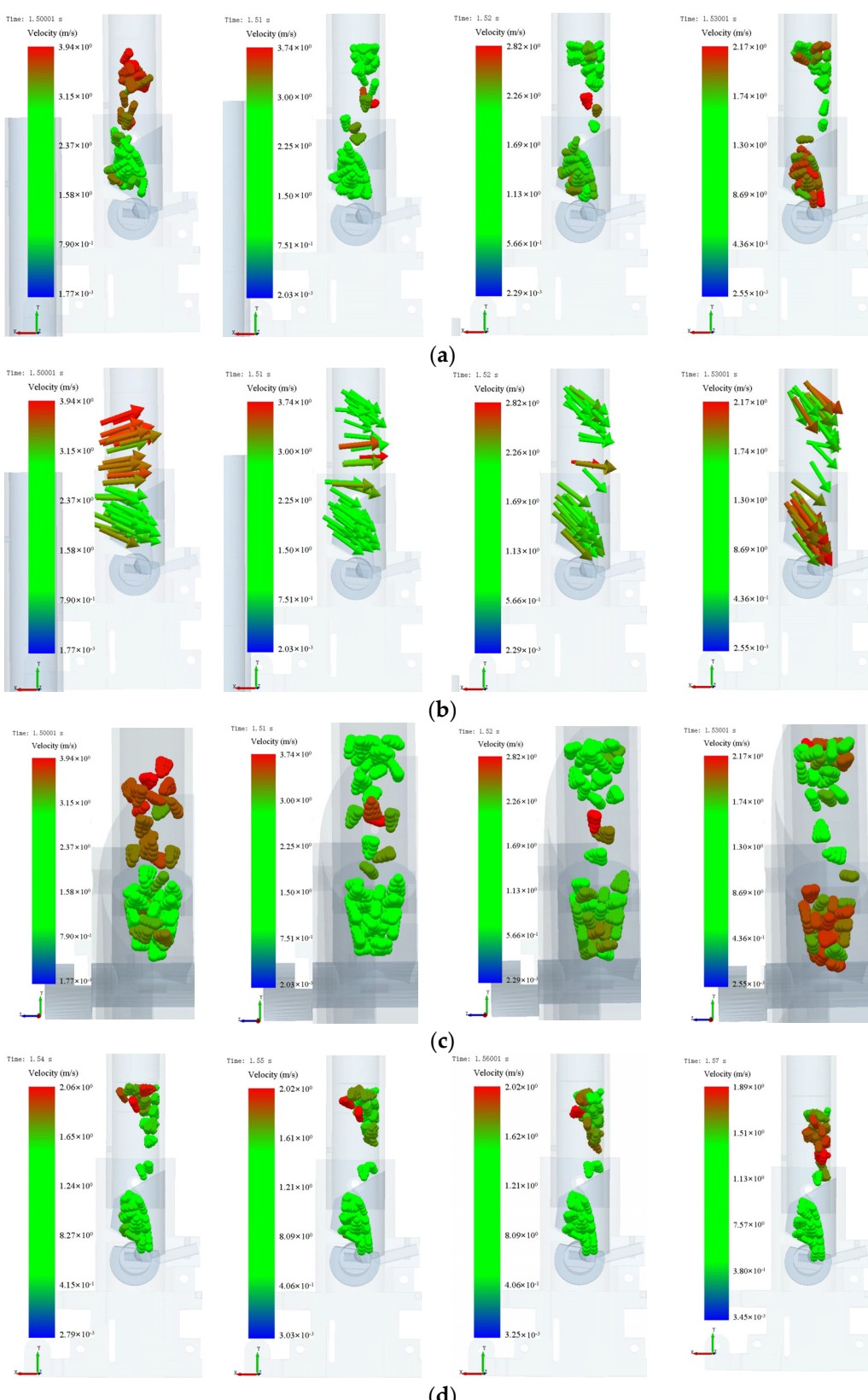

**Figure 20.** *Cont.*

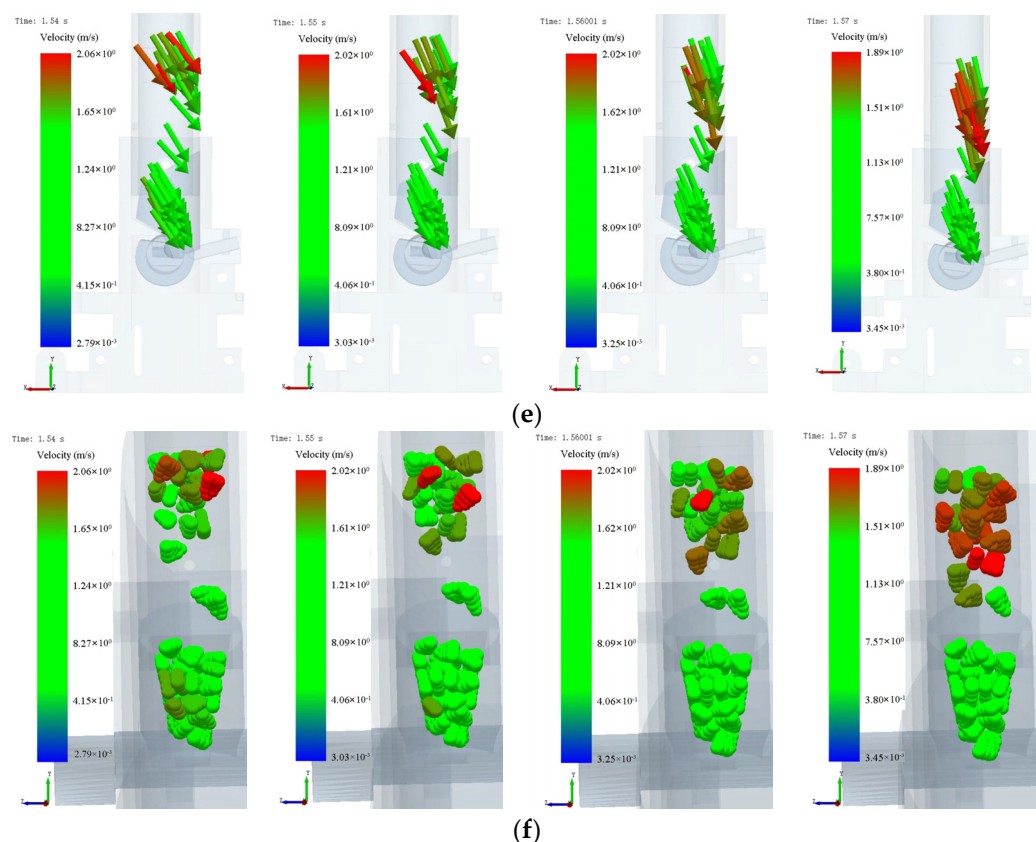

**Figure 20.** Movement process of maize in the seed tube with inverted hook added for guiding seed. (**a**) Default display of left view. (**b**) Vector display of left view. (**c**) Default display of front view. (**d**) Default display of left view. (**e**) Vector display of left view. (**f**) Default display of front view.

### 3.8. Simulation and Experimental Verification

From the above mechanism of interaction between maize seeds and cavity seeders, the effect of the seeder's movement on seed filling has been determined. It is necessary to verify the seeding performance under certain conditions of the shaped groove structure size in the next step. The diameter of the seed taking wheel was 22 mm, the included angle of the inner groove line was 110°, the width of the shaped groove was 11.79 mm, the height was 7.23 mm, the length was 14.67 mm, the opening angle was 15°, and the maximum diameter of the spherical hole was 5.5 mm. The shaped groove was made of high-precision 3D printing. The maize trajectory line during seeding is seen in Figure 21.

Based on simulation experiments, the single grain rate was 75%, the reseeding rate was 25%, and the missed seed rate was 0%. Additionally, a field trial was conducted at Taohe Tractor Factory, Lintao County, Gansu Province, China, on 11 March 2021. The experiment field was oriented east–west with pre-rotational plowing and leveling. It was a sandy loam soil. The soil water content ranged from 11.7% to 13.6%, and the average soil firmness measured with a TJSD-B firmness instrument at 45 mm depth was 137 kg/cm$^2$ (Beijing Jingcheng Huatai Instrument Co., Ltd., Beijing, China). The maize material was Longdan No. 339. The mass of one thousand grains was 371 g, and the average size of the seeds was 14.2 mm by 10.8 mm by 7.3 mm. We used a self-propelled cavity planter of type 2BZ-2 with vertical insertion at a forward speed of 0.5 m/s. The prototype is illustrated in Figure 22.

Before optimization, the best seeding result of the cavity planter was a seed number qualification of 75% and the empty cavity rate was 15%. After optimization, the results of the experiment indicated that the average seed number qualification and the empty cavity rate were 2% and 91.30%, respectively. Thus, the seed number qualification increased by 17.85%, and the empty cavity rate decreased by 86.67%. Meanwhile, it was found that the drive of the cavity planter operated smoothly during the experiment. Additionally,

the forward speed compensation mechanism compensated for the forward speed of the machine without jamming or slipping. Experimental results were in accordance with the design and agronomic requirements [27].

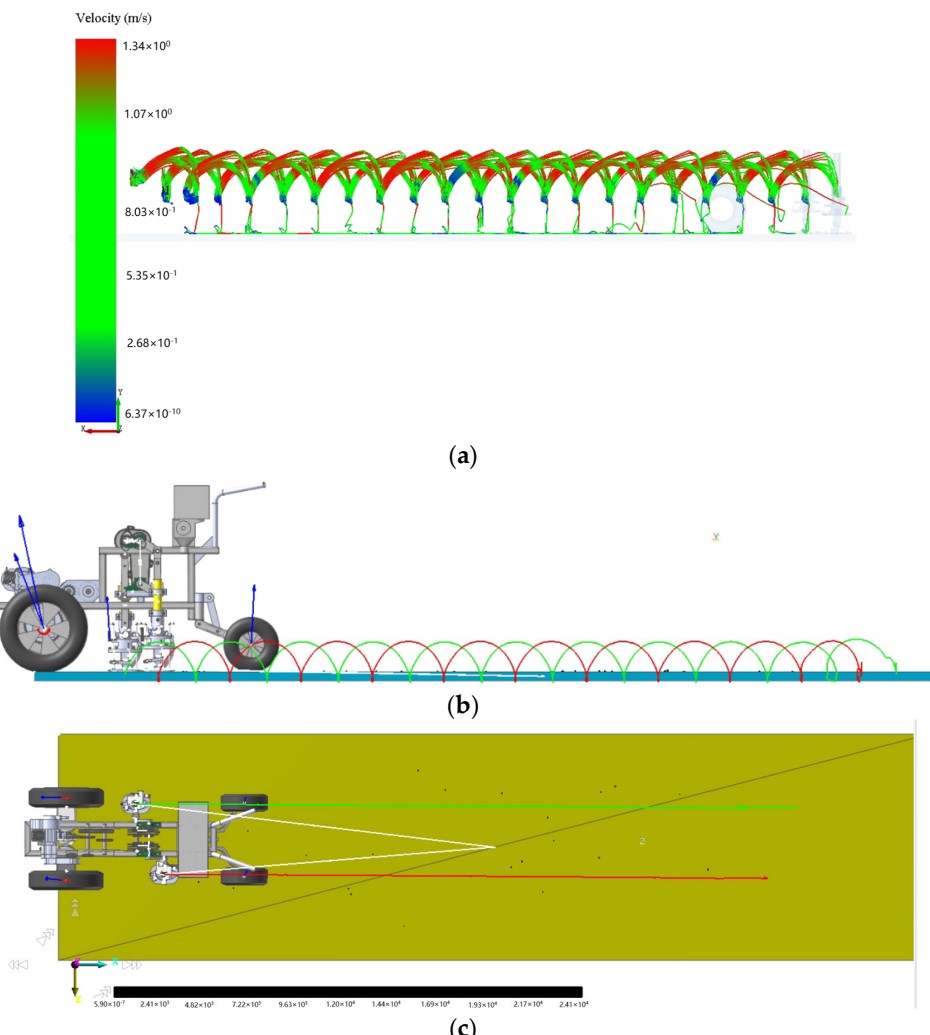

(a)

(b)

(c)

**Figure 21.** Simulation seeding performance of the cavity planter with vertical insertion. (**a**) Seed movement trajectory. (**b**) Trajectory of the cavity seeder (main view). (**c**) Trajectory of the cavity planter (top view).

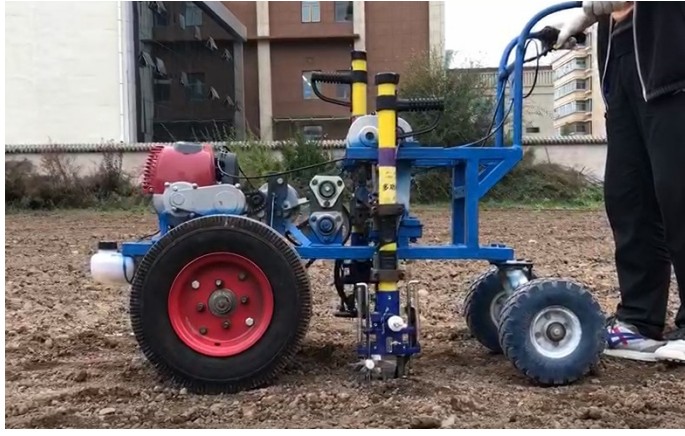

**Figure 22.** 2BZ-2 type cavity planter with vertical insertion.

## 4. Conclusions

A simulation experiment was conducted using the DEM-MBD coupling method to study the soil disturbance by the cavity planter and the entry resistance. Additionally, the effect of maize shape and the cavity seeder motion on the seed number qualification and the empty cavity rate were studied in detail.

(1) The results of the simulation show that soil type and water content influence the trajectory of the cavity seeder. The slip rate of the cavity planter on sandy loam is significantly higher than that of sandy soils at the same water content. As the speed of the traction wheel decreases, the trajectory of the cavity planter gradually moves backward. As the ratio decreases, the traction wheel speed increases, and the motion trajectory gradually advances. The cavity planter causes soil disturbance due to the different types of soil. It is recommended to moderately disturb the soil above the intersection of trajectory to enlarge the hole seeder and increase air circulation.

(2) During the cavity seeding machine's operation, the resistance of the cavity planter basically increased with the rise in the water content of sandy soil and sandy loam. It was found that the entry resistance of sandy loam is significantly higher than that of sandy soil.

(3) As the contact height between maize and the inner wall was large, this did not facilitate seed filling. After two improvements to the inverted hook structure in the seed tube, the maize model above the inverted hook moved at a faster speed and had an uneven distribution. The inverted hook for guiding seeds enhanced the seed filling. Finally, the seeds' seeding performance was verified under certain conditions of the shaped groove structure size with an angle of the inner groove line of 110°, the width of the shaped groove of 11.79 mm, the height of 7.23 mm, the length of 14.67 mm, the opening angle of 15°, and the maximum diameter of the spherical hole of 5.5 mm. The results showed that the empty cavity rate was 25%, and the seed number qualification was 75%, while the average empty cavity rate and the seed number qualification were 2.0% and 91.3%, respectively. The improved structure meets design and agronomic requirements.

**Author Contributions:** Conceptualization, W.Z.; methodology, L.S.; Recurdyn software, C.H.; formal analysis, G.R.; investigation, Z.W. and J.G. All authors have read and agreed to the published version of the manuscript.

**Funding:** This research was funded by the Excellent Ph.D. Dissertation of Gansu Agricultural University: YB2020003; The National Natural Science Foundation of China grant number: 52065004, 52165028.

**Institutional Review Board Statement:** Not applicable.

**Informed Consent Statement:** Not applicable.

**Conflicts of Interest:** The authors declare no conflict of interest.

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
