# Peer review of "Study on the Intercropping Mechanism and Seeding Improvement of the Cavity Planter with Vertical Insertion Using DEM-MBD Coupling Method"

_agriculture, doi:10.3390/agriculture12101567_

Round 1

Reviewer 1 Report

The core of this paper is to use DEM-MBD coupling technology to carry out the simulation test of soil disturbance and entry resistance of cavity seeder, which takes into account a variety of influencing factors in the process of sowing (soil type, seed shape, soil water content, etc.), and optimize the design of cavity seeder and carry out the practice. The experimental results show that the qualified rate of the cavity planter after optimized design is higher.

To sum up, this paper has clear logic and detailed content. The experimental results have certain guiding significance for the development of seed drill with cavity.

1. Some spelling mistakes in this article are as follows:

(1) DEM in 17 lines of the abstract is mistakenly written as EDM.

(2) In line 455 of the conclusion, DEM is mistakenly written as EDM.

2. In this paper, some figures are annotated and marked incorrectly as follows:

(1) The caption in line 117 is not capitalized.

(2) The first letter is not capitalized in Figure 2d.

(3) The non-English label appears in Figure 9a and Figure 9b.

(4) The non-English label appears in Figure 10a/b.

(5) Figure 11b shows the non-English label.

(6) In Figure 14a/b/c, the first letter is not capitalized.

(7) Figure20a has too much indentation.

3. In order to make the chart in this paper clearer and easier for readers to read, it is suggested to modify it as follows:

(1) The title of Table 3 should be placed on the same page as the content suggestion.

(2) The caption of Figure 8 is suggested to be placed on the same page as the figure.

4. The research core of this paper is to improve the design of cavity planters based on DEM-MBD coupling, but the introduction part contains a lot of research process about soil statics. It is suggested to simplify and refine the introduction part so that readers can quickly capture the innovation of this paper.

5. In line 73~76 of the introduction, the stress status between the planter and the soil is analyzed. It is suggested to add a stress analysis chart, so that readers can directly see a stress status of it.

6. Figure 18. It is recommended to add a 3D model to show the readers the actual model of your improved barb.

7. Lines 448 to 453 in this paper illustrate the results of the field experiment of the improved seed drill, but there is no comparison. How much is the qualified rate of seeding before the improvement of the seeder, and how much is the optimization coefficient?

Author Response

Many thanks to the reviewers for their valuable comments, and the paper has been revised line by line and highlighted in red.

  1. Some spelling mistakes in this article are as follows:

(1) DEM in 17 lines of the abstract is mistakenly written as EDM.

Reply:EDM has been changed to DEM.

(2) In line 455 of the conclusion, DEM is mistakenly written as EDM.

Reply:EDM has been changed to DEM.

  1. In this paper, some figures are annotated and marked incorrectly as follows:

(1) The caption in line 117 is not capitalized.

Reply:The caption in line 117 has been capitalized.

(2) The first letter is not capitalized in Figure 2d.

Reply:The caption in Fig.2d has been capitalized.

(3) The non-English label appears in Figure 9a and Figure 9b.

Reply: Changes have been made to the x-axis names in Figure 9.

(4) The non-English label appears in Figure 10a/b.

Reply: Changes have been made to the x-axis names in Figure 10.

(5) Figure 11b shows the non-English label.

Reply: Changes have been made to the x-axis names in Figure 11b.

(6) In Figure 14a/b/c, the first letter is not capitalized.

Reply:The caption in Fig.14a, b, c has been capitalized.

(7) Figure20a has too much indentation.

Reply: Indentations have been modified.

  1. In order to make the chart in this paper clearer and easier for readers to read, it is suggested to modify it as follows:

(1) The title of Table 3 should be placed on the same page as the content suggestion.

Reply: Table 3 has been adjusted for completeness

(2) The caption of Figure 8 is suggested to be placed on the same page as the figure.

Reply: Figure 8 has been adjusted for completeness

  1. The research core of this paper is to improve the design of cavity planters based on DEM-MBD coupling, but the introduction part contains a lot of research process about soil statics. It is suggested to simplify and refine the introduction part so that readers can quickly capture the innovation of this paper.

Reply: I have revised and simplified the introduction in line with the theme of the thesis.

  1. In line 73~76 of the introduction, the stress status between the planter and the soil is analyzed. It is suggested to add a stress analysis chart, so that readers can directly see a stress status of it.

Reply: Addition of duckbill and soil stress changes during duckbill insertion into the soil in section 3.5.

  1. Figure 18. It is recommended to add a 3D model to show the readers the actual model of your improved barb.

Reply: I have provided 3D model.

  1. Lines 448 to 453 in this paper illustrate the results of the field experiment of the improved seed drill, but there is no comparison. How much is the qualified rate of seeding before the improvement of the seeder, and how much is the optimization coefficient?

Reply: We have added before and after optimization results comparation in section 3.8.

Reviewer 2 Report

This paper addresses the practical problems that arise during the development of direct hole seeders, and with the help of EDM-MBD coupling technology for direct hole seeding simulation, it investigates the disturbance of the hole seeder on different types and moisture content soils, and the change in the entry resistance of the hole-former, as well as the effect of three types of maize on the seed discharge performance, further revealing the details of the hole seeding mechanism occurring. Some reference is provided for improving the quality of direct hole sowing. Specific modifications are suggested as follows.

1. What is the method used to ensure the dimensional accuracy of 11.79 mm for the width of the shaped groove of the seed extraction wheel and 7.23 mm for the height ensured by which processing method?

2. The structural features of the shaped groove of the seed extraction wheel are not described in the Paper.

3. It is suggested that DEM be added to the keywords.

4. Please add to section 2.1 whether there is any involvement of the quantitative seed discharge system in the interactions of the seeder.

5. Please add the country and version of the RecurDyn software in section 2.1, and similarly for the EDEM software.

6. Please add to the paper how the seeder can be flexibly steered on the ground and suggest that the steering mechanism be implemented.

7. Please check the subscripts of the yield strength values in Table 1, and check in the full paper.

8. The dimension for the 3 types of maize seeds should be supplied.

9. Chinese characters appear on the x-axis in Fig. 9 and Fig. 10, please modify.

10. The forward speed in section 3.2 are accurate to three decimal places, it is suggested that they be standardized to two decimal places in the full paper.

11. In section 3.6, in addition to the influence of different types of maize on seeding performance, the movement of the seeder was also found to be a major factor and it is suggested that this be analyzed further in the paper.

12. Check if the translation of Tahoe Tractor Factory in line 439 is wrong.

13. Add information on the instruments used to measure soil firmness.

14. Suggest additional discussion in chapter 4 to further analyze the reasons for the simulation and experiment results.

Author Response

Many thanks to the reviewers for their valuable comments, and the paper has been revised line by line and highlighted in red.

  1. What is the method used to ensure the dimensional accuracy of 11.79 mm for the width of the shaped groove of the seed extraction wheel and 7.23 mm for the height ensured by which processing method?

Reply: The shaped groove is made of high-precision 3D printing.

  1. The structural features of the shaped groove of the seed extraction wheel are not described in the Paper.

Reply: We have supplied the shaped groove and the structural parameters in Fig.1. In addition,

specific values have been supplemented in Section 3.8.

  1. It is suggested that DEM be added to the keywords.

Reply: We have add “DEM” in keywords.

  1. Please add to section 2.1 whether there is any involvement of the quantitative seed discharge system in the interactions of the seeder.

Reply: As the cavity seeder is the main part of the seeder, which participates in the interaction process of corn and seeder, we have explained in Section 2.1.

  1. Please add the country and version of the RecurDyn software in section 2.1, and similarly for the EDEM software.

Reply: we have supplied in Section 2.1.

  1. Please add to the paper how the seeder can be flexibly steered on the ground and suggest that the steering mechanism be implemented.

Reply: Ground drive system include differential mechanism. The cavity seeder can turn flex-ibly on the ground. The traction wheel can be turned at different speeds by turning the handle. we have explained in section 2.1.

  1. Please check the subscripts of the yield strength values in Table 1, and check in the full paper.

Reply: We have revised the subscripts in the full paper.

  1. The dimension for the 3 types of maize seeds should be supplied.

Reply: We have supplied the picture with dimension for the maize seeds.

  1. Chinese characters appear on the x-axis in Fig. 9 and Fig. 10, please modify.

Reply: Changes have been made to the x-axis names in Fig.9 and Fig.10.

  1. The forward speed in section 3.2 are accurate to three decimal places, it is suggested that they be standardized to two decimal places in the full paper.

Reply: We have be standardized to two decimal places in the full paper.

  1. In section 3.6, in addition to the influence of different types of maize on seeding performance, the movement of the seeder was also found to be a major factor and it is suggested that this be analyzed further in the paper.

Reply: We have added analysis and suggestion about seed shape effect in section 3.6.

  1. Check if the translation of Tahoe Tractor Factory in line 439 is wrong.

Reply: We have revised Tahoe Tractor Factory as Taohe Tractor Factory.

  1. Add information on the instruments used to measure soil firmness.

Reply: We have supplied soil firmness instruments information in section 3.8.

  1. Suggest additional discussion in chapter 4 to further analyze the reasons for the simulation and experiment results.

Reply: We have added before and after optimization results comparation in section 3.8.

Round 2

Reviewer 1 Report

Compared with before the modification, this paper has a significant improvement, the argument is concise and clear, the logic is clear, and the format is standardized. The modification of the introduction part is more dazzling, readers can quickly capture the innovation of the paper and understand the core point of this paper.

The main modification highlights of this paper are as follows:

(1) The introduction explains the working process of the cavity seeder, and then explains the core thesis of this paper.

(2) In Section 3.5, the change of soil stress at the entrance of the sand loam hole seeder is added, so that readers can intuitively see the stress of the hole seeder.

(3) The reason for mixing some of the seeds together is clearly explained in Section 3.6.

(4) Figure 19 added the improved 3D model diagram of inverted hook.

(5) In Section 3.8, the qualified rate and hole rate of the cavity planter before optimization are added, which has a significant optimization improvement compared with the improved planter.

However, there are still some format problems as follows:

(1) The first letter of the caption in line 339 and line 341 is not capitalized.

(2) There are missing brackets in A, B and C after Figure 15 in line 369, 371 and 373.

Author Response

Dear reviewer

Many thanks to the reviewers for their valuable and accurate guidance on revisions.

The specific amendments as

(1) The first letter of the caption in line 339 and line 341 is not capitalized.

Reply:I have changed the initials in lines 339 and 341 to capital letters.

(2) There are missing brackets in A, B and C after Figure 15 in line 369, 371 and 373.

Reply:I have added brackets after Figure 15 on lines 369,371 and 373.

We would like to thank the reviewers and editors for their hard work.
